# Grow Your Own School Mental Health Specialists: A Policy Pilot to Address Behavioral Health Workforce Shortages in Schools

**DOI:** 10.3390/bs14090813

**Published:** 2024-09-13

**Authors:** Samantha M. Bates, Dawn Anderson-Butcher, Tyler Wolfe, Chris Ondrus, Sean Delaney, John Marschhausen, Olivia McAulay, Katie Klakos

**Affiliations:** 1Community and Youth Collaborative Institute, College of Social Work, The Ohio State University, 101 Bricker Hall, 190 N Oval Mall, Columbus, OH 43210, USA; anderson-butcher.1@osu.edu (D.A.-B.); delaney.63@osu.edu (S.D.); mcaulay.2@osu.edu (O.M.); klakos.2@osu.edu (K.K.); 2Dublin City School District, 5175 Emerald Pkwy, Dublin, OH 43017, USA; wolfe_tyler@dublinschools.net (T.W.); ondrus_chris@dublinschools.net (C.O.); marschhausen_john@dublinschools.net (J.M.); 3College of Education and Human Ecology, The Ohio State University, 127 Arps Hall, 1945 North High Street, Columbus, OH 43210, USA

**Keywords:** grow your own, workforce development, school social work, behavioral health-education system partnerships

## Abstract

The capacity of schools to address behavioral health concerns presents an emerging challenge, exacerbated by major shortages in the workforce. Schools across the U.S. are struggling to hire licensed behavioral health professionals, with additional barriers encountered when seeking to hire practitioners with experience in educational settings. In 2023, a school district in the suburbs of Columbus, Ohio, partnered with The Ohio State University to launch a “grow your own” policy pilot. The priorities focused on addressing workforce shortages and leveraging the experiences of current teachers/staff to support growing needs related to student mental health and well-being. More specifically, the district utilized COVID-19 relief funds to recruit, train, and transition 25 teachers/staff into school mental health positions by underwriting the costs of each professional’s Master of Social Work (MSW) degree. Here, we (a) describe the district–university partnership and the processes guiding the implementation of the “grow your own” model, (b) distill preliminary findings about district needs regarding behavioral health, and (c) explore facilitators, barriers, and outcomes associated with learning among participants in the program. The findings from a district-wide staff survey indicated a high level of need for individual counseling, crisis intervention, and small group interventions. Additionally, qualitative interviews revealed that learning among the program’s participants was facilitated by effective classroom strategies and specific learning experiences integral to the program’s design. These facilitators supported key learning outcomes, including general social work knowledge, self-awareness, and therapeutic skills that are foundational for engaging with students, parents/families, teachers/staff, and the broader school community. This innovative policy pilot and training model demonstrate how universities and local educational agencies can partner to address workforce development challenges at the intersection of behavioral health and education.

## 1. Introduction

Schools and communities worldwide have witnessed an increased need for child and adolescent behavioral health services [1,2]. Internationally, Racine et al. published a meta-analysis of 29 studies that found that the prevalence of childhood anxiety and depression has doubled from pre-pandemic estimates [2]. Trends are also alarming in the United States (U.S.), where the Centers for Disease Control and Prevention (CDC) reported a 40% increase in mental and behavioral health symptomology among young people in the past decade [3]. Schools are uniquely positioned to address these growing needs, yet their capacity to do so remains challenging [1]. One substantial factor influencing a coordinated response to the behavioral health crisis in schools is the nationwide shortage of therapists, social workers, behavioral health counselors, psychologists, and psychiatrists [4]. This challenge is further exacerbated when seeking to hire professionals with experience in educational settings.

To address workforce shortages and respond to behavioral health needs in schools, teacher education, school psychology programs, and other disciplines have sought to recruit, train, and retain professionals by developing “grow your own” training programs [5,6]. Historically, these programs have recruited high-school and college students, as well as adults from schools or communities, into educational pathways that help them obtain specific licenses (e.g., special education, teaching). The success of these models is evident in their high retention rates. Guha et al. found that retention rates among teacher residency programs, where teachers are trained and hired in designated high-need schools, ranged from 70% to 80% after five years for program graduates [7]. Program developers have underscored the critical importance of strong university–district partnerships alongside financial incentives such as loan forgiveness and retention bonuses to ensure the success of these interventions [7,8]. The potential of such programs is promising.

To date, similar “grow-your-own” models have not been replicated with social workers trained to work in educational settings, and no programs, to our knowledge, have recruited current school staff to re-engage in social work coursework to address behavioral health workforce shortages [5,6,7,9]. Notably, a Master’s in Social Work (MSW) with emphases on school social work (SSW) practice can enhance the skills, knowledge, and competencies of graduates to deliver behavioral health services in schools, especially those serving highly impacted students with diverse learning needs [10,11]. Given opportunities to implement a “grow your own” model with social workers and the timeliness of this approach, the current article details a university–district partnership in Ohio supporting the transition of 25 mid-career teachers, staff, and paraprofessionals into SSW (e.g., mental health specialist) roles by underwriting the costs of MSW degrees and preparing participants to address growing behavioral mental health needs across the district. This article describes the design and implementation of the “grow your own” model, explores emergent behavioral health needs in the district, and examines participant perceptions of facilitators of and barriers to learning. Learning outcomes among program participants also are examined to better understand what knowledge, skills, and competencies are gained by completing one year of the MSW program with an emphasis on SSW practice. The results frame a discussion regarding lessons learned and future opportunities to replicate “grow your own” programs to address behavioral health workforce shortages. 

## 2. Literature Review

### 2.1. Behavioral Health Needs and Workforce Shortages in Schools

In the U.S., behavioral health concerns impairing youth in schools are at historical highs and continue to rise [3]. Behavioral health encompasses a broad range of factors that impact both physical and mental well-being, including mental health (e.g., anxiety, depression), lifestyle and health behaviors, addictions, substance misuse and abuse, stressful and crisis situations, and learned emotional responses [3,12,13]. Before the pandemic, national survey data indicated that adolescent depression had increased from 8.3% in 2011 to 12.9% in 2016 [14]. During the pandemic, emergency department visits for suicidal ideation surged, with children and adolescents increasingly engaging in suicidal attempts and self-harm behaviors [15]. Scholars also found that after the first year of lockdowns, 20% to 25% of youth worldwide were experiencing clinically significant levels of depression or anxiety [2]. Trends indicate an overwhelming rise in behavioral health needs during the pandemic, which became evident in schools as they reopened.

As lockdowns were lifted in schools across the country, the pandemic’s impact on behavioral health outcomes among youth has been of increasing concern. For example, in addition to school district leaders reporting that students had fallen behind in their social–emotional development [16], other studies found that 48% of students perceived a decline in their social–emotional competencies upon returning to school, with those previously receiving school-based support services being the most affected [17]. These trends prompted the Office of the U.S. Surgeon General to issue an advisory statement urging communities to devise creative solutions to ensure every child had access to high-quality, affordable, culturally competent behavioral health care, especially in educational settings [18]. The statement recommended that the U.S. take action to expand and support the education workforce with a focus on youth behavioral health. However, this call to action was matched by national and state shortages of current and future behavioral health professionals, further challenging schools to finance, manage, and expand the provision of accessible and effective behavioral health services.

Notably, behavioral health workforce shortages were already an issue before the pandemic, but supply and demand issues have intensified and are expected to persist. The NCHW reported that the recruitment and retention of behavioral health professionals were concerns pre-pandemic due to secondary trauma, high caseloads, and demanding work [4]. However, significant shortages in counselors, social workers, marriage and family therapists, psychologists, and psychiatrists are projected through to 2036. Similarly, schools were severely understaffed even before the pandemic. For example, a report from the Hopeful Futures Campaign published in 2022 found that 48 states did not meet recommended school counselor-to-student ratios, 49 states failed to meet recommended school psychologist-to-student ratios, and no states met recommended social worker-to-student ratios [19]. Hence, schools nationwide were under-resourced before, during, and projected to be after the pandemic and currently face challenges in funding positions, hiring qualified practitioners, and retaining current staff [4,20].

### 2.2. Solutions: Residency and “Grow Your Own” Programs

One solution to workforce shortages in professions such as teaching, educational leadership, and school psychology has been to develop residency or “grow your own” models. These models aim to recruit pre-professionals, community members, and students to staff hard-to-fill school positions [6,7]. Typically, these programs operate as partnerships among a university and a school district, region, or state entity (e.g., the Department of Education) and are funded through federal or state resources. Strategies aim to support the successful recruitment, training, and retention of students in specific areas of need (e.g., psychology, special education, etc.), who are later hired in high-need schools. Key components of the programs include identifying students who already live and work in the communities where schools are located and offering financial incentives to participants [7]. However, recruiting individuals with backgrounds or interests in education can prove challenging, especially if costs or travel for coursework are barriers to program entry [7]. Furthermore, implementing “grow your own” approaches in the behavioral health sector has most commonly been funded through regional or state initiatives, and the understanding of how local entities can replicate these models remains relatively understudied [6,21,22]. 

The benefits of potentially implementing “grow your own” programs are noted in prior studies. Through “grow your own” programs, students gain training and insight into the culture of the schools where they will work. Further, field experiences are integrated into the coursework, and embedded supervision supports scaffolded learning and growth as students approach graduation [6]. Many programs also offer financial incentives, such as loan forgiveness, lower tuition fees, retention bonuses, and tuition reimbursement to encourage participation and reduce financial burdens [6,8]. In fact, financial incentives have been shown to improve retention outcomes [8]. However, the aforementioned options often fail to alleviate students’ financial burdens. Perhaps fully funding tuition and program costs would contribute more substantially to positive retention outcomes, both within the preparation program and in job placements post-graduation. 

Nationwide, school leaders, scholars, and practitioners are witnessing increased support for these programs, evidenced by Behavioral Health Service Demonstration grants and state-level funding for workforce development in behavioral health [21,22]. Numerous articles describe programs funded at the federal or state level for school psychologists, teachers, and school administrators, but not for social workers [6,7]. Consequently, opportunities exist to explore how models can be designed specifically for social workers, particularly those trained to work in educational settings as SSWers, sometimes referred to as mental health specialists. SSWers are uniquely positioned to address escalating behavioral health issues, especially given their training in direct clinical practice and macro-level functions to connect community-based resources to schools through expanded partnership models [23]. 

Importantly, the National School Social Work Association of America has developed a School Social Work Practice Model that outlines the roles of SSWers in providing evidence-based education, behavioral supports, and behavioral health services, promoting a school climate conducive to student learning and teaching excellence, and maximizing access to school- and community-based resources [24]. SSWers are equipped with advanced knowledge and skills in these three critical areas, and develop specialized skill sets focused on addressing students’ social and emotional needs. Additionally, SSWers are uniquely prepared to develop school–family–community partnerships, address non-academic barriers to learning [23], and coordinate services and resources across multiple sectors to address broader socio-political issues [25,26]. With growing mental health and concerns with other challenges among young people today, preparing past educators to become SSWers may prove to be a viable approach for addressing the growing behavioral health challenges that are currently evident. 

### 2.3. Current Study

This article describes a “grow your own” model developed and implemented through a district–university partnership in Ohio. The partnership aimed to increase the number of qualified behavioral health service providers (e.g., mental health specialists) offering school-based services to students with demonstrated need. The school district elected to address this need using COVID-19 relief funds to underwrite the full tuition costs of a two-year MSW degree for 25 teachers, paraprofessionals, and staff currently employed by the school district. The program was designed with design elements prioritizing local oversight and eliminating financial and travel burdens for students as part of an implementation plan to retain all 25 students throughout the program. This cohort, referred to hereafter as the “cadre”, are working toward completing their MSW degrees and obtaining an Ohio SSW license. After one year of program implementation and retention of all 25 cadre members, researchers examined the following exploratory research questions: (a) What are the underlying needs of the district concerning behavioral health supports? (b) What are student perceptions of programmatic and interpersonal facilitators and barriers influencing their learning and retention? (c) What learning outcomes are achieved after one year of implementing this “grow your own” program?

## 3. “Grow Your Own” Program Implementation and Design 

This article describes the “grow your own” model with attention to the program’s implementation and design. Preliminary findings about the district’s need for behavioral health support are described, and facilitators, barriers, and outcomes associated with learning upon the cadre’s completion of one year of the program are described. 

### 3.1. Context

The current project took place in the Dublin City Schools (DCS) district, located in the suburbs of Columbus, Ohio. Behavioral health workforce shortages are notable in Ohio. Indeed, workforce data from the Ohio Department of Mental Health and Addiction Services (OhioMHAS) indicate a 353% increase in the demand for behavioral health treatment between 2013 and 2019, with an average annual increase of 29% [27]. Recently, OhioMHAS also predicted an annual increase in statewide demand of 5.6% per year over the next decade [28]. Furthermore, Ohio has a deficit of 1,337 behavioral health counselors and 2895 social workers to meet projected demands in 2030 [27]. Locally, DCS is the ninth largest district in Ohio and encompasses 47 square miles. The district comprises 24 schools and continues to grow, serving approximately 16,000 students (with over 70 languages spoken) and employing 2,500 staff [28]. Currently, the district employs 18 mental health specialists who hold SSW or counseling licenses, resulting in a student-to-behavioral-health-provider ratio of approximately 1:888 (compared to the recommended 1:250 by SSWAA). In the year following the pandemic, the district recorded 243 incidents of screenings for suicidal ideation, indicating a significant increase in behavioral-health-related concerns unprecedented in the DCS. 

### 3.2. District–University Partnership

Since 2019, DCS has partnered with The Ohio State University (OSU), the state’s flagship land-grant university, to offer internships to social work students pursuing their MSW degrees. The OSU College of Social Work (CSW) is ranked 12th in the country amongst social work programs, according to U.S. News & World Report [29]. Specifically, the Community and Youth Collaborative Institute (CAYCI) within the CSW at OSU has overseen the successful training of emerging SSWers through practicum experiences, helping the district fill staff vacancies using this training model. However, when the district sought to increase its staff to place a provider in each building following the pandemic, school and university leaders experienced firsthand the effects of the behavioral health workforce shortage and the challenges of identifying qualified practitioners experienced in school settings. In 2022, the district superintendent proposed using COVID-19 relief funds to launch a pilot “grow your own” policy pilot in partnership with OSU-CAYCI. The pre-established relationship between the university and the school district helped to actualize this idea. 

Beginning in the fall of 2022, partners from OSU-CAYCI and DCS began implementing this policy pilot by collaborating to recruit, screen, enroll, and train 25 current teachers, paraprofessionals, or school staff interested in pursuing an MSW degree. Upon graduation, these “teacher to social workers” would transition the following year into school mental health specialist positions upon graduation. Notably, at the time of writing, these students had completed approximately half of the required coursework and other requirements for their MSW degrees. The following section outlines the processes used to identify current individuals for the policy pilot and describes the overall design of the “grow your own” program model. 

### 3.3. Participant Recruitment

In the fall of 2022, DCS leaders began recruiting participants by sending email communications to all district staff, including teachers, paraprofessionals, and professionals in student support roles (such as occupational therapists and physical therapists). Detailed information about the program, its objectives, and the benefits was outlined to ensure potential participants had clarity in relation to the overall policy pilot experience. The DCS organized a series of information sessions to further engage and inform interested individuals. These sessions offered a platform for potential participants to ask questions, gain deeper insights into the program requirements, and understand the commitment involved. This combined approach of informative emails followed by interactive sessions successfully generated interest among 140 prospective participants across the district.

### 3.4. District-Specific Screening, Application, and Selection

DCS and university leaders collaborated to develop a district-specific screening application. Based on readiness-for-change studies used in school improvement planning research [25], the application required participants to respond to several questions detailing their interest in the program (i.e., value), their motivation to transition into a school mental health role (motivation), past experiences or education (i.e., ability), and any facilitators and/or potential challenges (i.e., resistance, circumstances, etc.) they anticipated if they were to be accepted. Additionally, participants were asked to submit a letter of support from a supervisor within the district. In total, 42 professionals across the district applied, indicating their intention and desire to participate in the preparation program. Once applications were submitted, the district partnered with university colleagues on a blind review process. External experts in school-based behavioral health served as reviewers, including university faculty from other institutions, current SSWers from other districts, and private practitioners specializing in practice with children and adolescents. Each application was reviewed using a detailed rubric by at least three reviewers. 

Additionally, the 42 candidates underwent interviews conducted by the district’s Director of Student Well-Being. During these interviews, candidates were asked questions about their interests, career aspirations, perceived challenges, and abilities (such as professionalism, experience, and interprofessional collaboration). Additionally, references were verified via phone calls and emails to candidates’ supervisors. Based on these two evaluative, selective methods, the top 25 candidates, as ranked by external reviewers and the Director of Student Well-Being, were invited to join the program as the “Dublin Teacher Cadre”. These prospective students then officially applied to the OSU MSW program and participated in the university and college’s traditional graduate school admissions review process. The final 25 cadre members’ demographics included one physical therapist, three intervention specialists, seven paraprofessionals, 11 general education teachers (including two art teachers), two school psychologists, and one instructional coach.

### 3.5. Program Design

Given the district’s strategy of using current staff to address behavioral health workforce shortages, special design considerations were warranted for the “grow your own” policy pilot (see Table 1). These considerations aimed to maintain the district’s workforce needs, prepare participants for transitioning out of their current roles, and allocate tuition funds for each participant. One first step was mapping the MSW program expectations across five semesters to facilitate tuition allocations and allow students to remain in their full-time roles during the first year, providing ample time to identify replacements for their positions. 

### 3.6. Degree and Specialization

The OSU MSW program is a 63-credit hour degree accredited by the Council on Social Work Education. At OSU, MSW students can pursue the SSW Licensure Program through a specialized curriculum. Completing this coursework makes graduates eligible for an SSW license from the Ohio Department of Education, enabling them to seek employment in public, private, and charter schools, as well as in community behavioral health agencies that provide services in schools. The specialized curriculum includes SSW courses and requires the completion of an advanced field practicum in a school setting.

### 3.7. Coursework and Practicum Experiences

Coursework was designed to create a hybrid experience, allowing students to work full-time while taking several key courses in person and online. Instructors with expertise in specific content areas and experience teaching adult learners were selected for this program. Notably, the MSW requires the completion of two practicum experiences (also referred to as internships) under the supervision of licensed social workers. The first practicum provides a foundational generalist experience, while the second focuses on advanced practice, often in the setting where the student intends to seek employment (e.g., a school, hospital, or social service agency).

The cadre completed their first practicum through a university-wide initiative, LiFEsports (www.lifesports.osu.edu (accessed on 1 July 2024)), a partnership led by leaders in the College of Social Work, the Department of Athletics, and the Department of Recreational Sports at OSU. LiFEsports is a sport-based positive youth development (PYD) program that serves approximately 800 underserved youths aged 6 to 14 annually. The program uses sports to teach social skills and address inequalities in access to sport, recreation, and play, primarily during the summer months. Over 80% of these youths identify as Black/African American and live at or below 200% of the poverty line. Youths are transported from local recreation centers by bus, and programming is offered on OSU’s campus, at three community recreation and park centers, and one local elementary school. The role of cadre members in this practicum was to support youth behavior, with a special emphasis on cultural competence, trauma-informed care, and PYD. 

The second practicum experience was designed to take place in DCS elementary, middle, or high schools. Here, the “grow your own” experience became evident as the 25 cadre members served as student interns in buildings across the district. The practicum followed a dyad structure, where students were paired with cohort members from a different school and spent one day a week observing and supporting service delivery in that partner school. Overall, the practicum experience was designed to give cadre experiences in traditional SSW practices, inclusive of individual and group counseling, assessment and intervention planning, linkage and referral, program design, implementation and evaluation, case management and crisis. 

### 3.8. Supervision and Additional Support

Practicum experiences were thoughtfully designed in collaboration with OSU-CAYCI and the Field Office in the CSW. A former SSW practitioner with over ten years of experience provided group and individual supervision to students as they completed their practicums at LiFEsports. In the second year, students were assigned task instructors (day-to-day managers) in their assigned DCS schools but continued to receive field instruction from the seasoned former SSW practitioner. Throughout the program, students in the cadre had a liaison from OSU-CAYCI who facilitated communication between DCS and OSU. The liaison assisted students with course enrollment, the transfer of graduate credits or transcripts, coordination of meetings for DCS and OSU-CAYCI leaders to resolve issues or discuss design decisions, and organizational tasks related to tuition, including coordinating communications among the graduate school, DCS, and student fee dashboards. 

Collectively, the policy pilot is an innovative design strategy for addressing growing behavioral mental health needs among children and adolescents today. The further examination of progress to date in relation to the need for, benefits of, and facilitators of learning here sheds light on the value of such “grow your own” programs for others. Next, the authors describe the methods and preliminary findings from the first year of cadre preparation.

## 4. Methods

This study explored initial perceptions of district needs related to behavioral health, demonstrating the value and importance of this work for the DCS school community. Additionally, scholars examined facilitators, barriers, and learning outcomes associated with the completion of year one of the program. 

### 4.1. Assessing District Perceptions of Behavioral Health Needs

#### 4.1.1. Recruitment

As the cadre engaged in the first year of their MSW program, district and university leaders partnered to implement a district-wide survey of staff to gather perceptions of behavioral health needs and the strength of the overall learning system. OSU researchers sent recruitment scripts to DCS administrators, who forwarded an online survey to all teachers/staff in their buildings in the spring of 2024. The survey included online informed consent information. The OSU Institutional Review Board approved all study procedures. 

#### 4.1.2. Sample 

In total, 596 district staff completed the survey (representing about 40% of all district employees). Of these participants, 77% identified as teachers, 16% as student support staff, 3% identified as administrators, 2% as paraprofessionals, and 2% as other school roles (e.g., classified, substitute teachers, teacher leaders/instructional coaches, etc.). Among those who completed the survey, 70% reported teaching in regular classrooms, 19% taught in special education classrooms, 2% taught in alternative settings, and 9% taught in other settings, such as art, gifted and talented, English language learning, physical education, and library settings. Overall, 30% worked in DCS for less than five years, 37% worked in DCS for between 6 and 15 years, and 36% worked in DCS for more than 16 years. 

#### 4.1.3. Measures

The district implemented the CAYCI School Experience Survey Teacher Version [30] to distill teacher and staff perspectives. The CAYCI School Experience Survey is an approved measure of the National Center on Safe Supportive Learning Environments school climate survey compendium [31]. In addition to the standard measures included in the teacher/staff version of the tool [32], several items were added to the measure to advance what is known about behavioral health needs in alignment with Weist et al.’s [33] call to revisit projections of the number of students needing resources or supports mapped within a multi-tiered system of supports model in schools. Nine items about the availability of and need for resources were co-developed with DCS leaders to assess resource availability and needs. Items were modeled and formatted in alignment with other school improvement and professional development needs assessments [34]. The stem of the items read, “In this building, there are enough resources to:”, and items reflected various levels of intervention, as mapped in the MTSS model, such as “screen students for mental health concerns”. Response options included: (a) Do Not Have, (b) Have and Do Not Need More, and (c) Have and Need More. 

#### 4.1.4. Quantitative Data Analysis

Data were downloaded from Qualtrics and then screened and cleaned in version 28 of the Statistical Package for Social Sciences (SPSS). Missing data ranged from 0.04% to 2.5% on various items. Due to small amounts of missing data, listwise deletion was utilized, and responses with complete data were included in the analysis. Frequencies on items were examined to distill staff perceptions of behavioral health needs in the district. 

### 4.2. Facilitators, Barriers, and Learning Outcomes 

#### 4.2.1. Recruitment and Sample

Once students were admitted to the MSW program, OSU researchers recruited cadre members to participate in a five-year study that would follow the participants during the two-year MSW program and for three years post-graduation when deployed in school mental health roles. Participants interested in the study signed written informed consent documents, allowing researchers to contact them at different time points to capture various data points. In the spring of 2024, cadre members enrolled in the study were recruited to participate in individual interviews with an OSU researcher about their experiences and learning in the program to date. At this time, cadre members had completed one practicum and 34 credit hours. In total, 19 of the 25 (76%) cadre members participated in the interviews. The average age of the participants was 46 years old (*SD* = 8.36). Participants also reported having worked an average of 14 (*SD* = 6) years in education. In total, 89% of the sample identified as female (11% male), 84% identified as White (5% two or more races; 11% chose not to answer), and 58% had a Master’s degree (32% had a Bachelor’s degree; 11% chose not to answer). 

#### 4.2.2. Semi-Structured Interviews 

Individual interviews were conducted with cadre members over Zoom. Each interview lasted approximately 1 h. The semi-structured interview guide was designed to gather perceptions of students’ facilitators of learning (e.g., “Talk to me about the design of the program. What did you like/dislike? What, if anything, has helped you be successful? How did you best learn? What learning opportunities stand out to you?”) and knowledge or skills gained to date (e.g., “Tell us what skills, knowledge, and competencies you’ve gained”). Cadre members were probed to share examples and details about their responses. 

#### 4.2.3. Qualitative Data Analysis

Data were transcribed and de-identified for analysis. Content analysis was used to interpret textual data by identifying patterns, themes, and subthemes within the content [35]. Content analysis was chosen because it allows researchers to systematically categorize textual data and identify recurring themes. Data analysis methods began by transferring data from text in Word to Excel. Next, one research team member coded and sorted the text into themes [36]. This element involved coding qualitative data into categories that represented similar meanings or concepts [36]. These themes were then grouped and organized into broader categories to provide a structured understanding of the data. Following this, an expert review was conducted to ensure theme validity, relevance, and comprehensiveness. The expert reviewer provided feedback and suggested modifications based on their knowledge and experience in the field. Additionally, a member check was performed, which involved sharing the results with one of the participants to verify the accuracy and resonance of the findings [37]. The participant provided insights and confirmed whether the identified themes accurately reflected the cadre’s experiences and perspectives. The expert review and member check enhanced the credibility and reliability of the analysis, ensuring that the final themes and subthemes accurately represented the data. 

## 5. Results

### 5.1. District Perceptions of Behavioral Health Needs 

Our first research question examined the underlying needs of the district concerning behavioral health supports. Overall, the greatest needs in the district included not having enough resources or supports to conduct home visits (50%), provide mental health literacy programming (34%), and engage in consultation for teachers (30%). Alternatively, staff reported having access to but needing more resources and supports in order to provide individual counseling (80%), support crisis intervention (75%), and implement small group interventions (70%). Table 2 overviews responses from staff regarding behavioral health needs and resource availability in the district. 

### 5.2. Facilitators and Barriers

Our second research question explored student perceptions of programmatic and interpersonal facilitators and barriers influencing their learning and retention. Interviews with cadre members pointed to several facilitators of learning and elements of the overall experience contributing most to their development as future SSWers. Table 3 highlights these facilitators. Key factors included several learning experiences and content that were instrumental to their growth, including learning about mental health (74%), content specific to SSW (79%), exploration of biases about race, class, and other social injustices (68%), and the ability to have real conversations in safe spaces (68%). Additionally, cadre members described instructional techniques and instructor qualities that facilitated learning, with the predominant qualities being their knowledge and expertise (58%) and flexibility (58%). They mentioned effective classroom instructional techniques, such as activities allowing for practical application (68%), presentations from social workers working in the field (53%), and group processing and reflection (50%). These experiences were amplified with courses offered in person, which were mentioned by 74% of the cadre members interviewed. This was due to the strong camaraderie among the cadre of learners, which was mentioned by 84% of those interviewed. Other helpful amplifiers of learning included the support, guidance, and structure provided by OSU (95%), whether or not they had prerequisite skills, knowledge, and experiences that were foundational to being a SSWer (74%), and the first-year field placement (53%). 

Additionally, themes emerged about challenges in and barriers to learning and the overall experience. The interviewed cadre members also highlighted several areas of improvement or issues with the program that needed to be addressed. The most salient themes included the demands and intensity of the MSW program (94%), missing family/personal life due to this intensity (84%), a lack of communication about what was coming next in the program (53%), and a disconnect and challenges with scheduling at the university (50%). Other themes involved the course content in relation to the online courses. The participants noted that the online courses involved busy work and the memorization of terms, as opposed to deep learning. Many indicated that they did not learn as much as on the in-person courses (79%). Other challenges and barriers included tensions among cadre members, which involved disrespectful classroom behaviors and chattiness (50%), outdated course materials (especially in the online setting; 42%), the desire for more mental health and less macro practice content (21%), a desire to learn more about community resources (21%), and a lack of diversity among the cadre members (21%). 

### 5.3. Learning Outcomes 

Our final research question explored student perceptions of the learning outcomes achieved after one year of engaging in the “grow your own” program. Specifically, the cadre members were asked what they had learned about skills, competencies, and knowledge during the first year of their experience and the Master’s program. Overwhelmingly, the participants mentioned how they increased their knowledge and understanding of the field of social work. Key areas regarding generalist practice concerned understanding the macro/mezzo forces and how the environment impacts individual opportunities and access (a theme that 79% of the interviewees mentioned). The participants also learned about the breadth and scope of the social work profession (53%) and the subdiscipline of SSW (53%). Additionally, 50% noted their growing confidence in their abilities as SSWers. The cadre members also reported how their self- and social awareness improved due to the learning experience. Specifically, 68% reported they were more aware of their biases and privilege, whereas 63% reported being more open-minded and/or having broadened their view of the world. Additional learning outcomes related to understanding and being able to search for evidence-based practices (50%) and finding community resources to support students and families (50%). Additionally, 58% reported increases in their skills in dealing with ethical dilemmas. Others learned ongoing lessons as students and social workers, such as how to “be a student again” (50%) and the importance of self-care (53%). Importantly, and in relation to the first foundational year of any MSW program, the interviewed cadres reported learning many therapeutic relationship-building skills central to social work practice and the helping profession. Examples included skills such as “going where the client is” (58%), listening skills (53%), patience (53%), and building rapport techniques (50%). The themes and subthemes in Table 4 represent key foundational skills taught in the first year of the program.

## 6. Discussion

This policy pilot, designated as a “grow your own” model, aimed to build the capacity of teachers and school staff in one local school district to obtain MSW degrees to address district needs and state and national behavioral health workforce shortages. The “grow your own” model designed, implemented, and described here is the first, to our knowledge, of a program designed specifically to train and prepare social workers for practice in education settings, and more specifically, to develop a cadre of SSWers [5,6,7,9]. The program demonstrates the unique opportunities for districts to leverage funds and current personnel to address shortages in the behavioral health workforce. Further, our study outlines systems-level innovations that showcase how universities and districts can create cohort programs that allow teachers/staff to work full-time and progress toward a graduate degree in social work. One year into the program, all 25 students (100%) have been retained and are progressing toward the timely completion of their degree, indicating success rates comparable to, if not better than, other “grow your own” models [7]. Researchers describe specific decisions, design elements, and resources, such as funding tuition for students, mapping the coursework over five semesters, engaging students in a practicum experience within a university-wide initiative supporting a population of highly vulnerable youths, and offering guided support throughout the program, which can be used to inform the replication of this model for other university–district partners.

Notably, the DCS had data that pointed to significant upticks in behavioral health needs during the COVID-19 pandemic that were matched by behavioral health shortages, both in Ohio and nationally [4,27]. Through the implementation of a district-wide survey, resource availability and needs were further substantiated as this project and study were underway, contributing to what is known about tiered needs related to behavioral health concerns in schools [33]. Overall, the district staff reported that the greatest gaps in support (e.g., do not have) existed at the Tier 1 and Tier 2 levels, in relation to providing mental health literacy, supporting classroom prevention planning, conducting home visits, and engaging in consultation for teachers. Without these formative upstream supports, the district is likely to experience an increase in the percentage of students requiring Tier 3 interventions [33]. Interestingly, the majority of the staff reported having support in several key Tier 2 and Tier 3 domains, yet there was an evident perception of the need for more resources to provide individual counseling, support crisis intervention, and implement small group interventions. Similarly, the teachers/staff reported having resources to screen for mental health concerns, but also desired more support in this area. 

The findings demonstrate baseline levels of needs in this district and point toward opportunities for this cadre to address gaps and meet behavioral health needs through numerous pathways. University instructors now have opportunities to leverage these data to focus efforts during year two of the program. These efforts should focus on sufficiently preparing practitioners to engage in prevention activities and support students with intensive intervention needs. Considered together, district and university leaders hope to see changes in these perceptions after the cadre enters into school mental health positions, potentially pointing toward broader long-term impacts of this learning cohort on district-level outcomes and behavioral health service resource acquisition. The data described in this study may provide evidence of changes in the traditional tiered system of support model, whereby more schools need practitioners prepared to deliver Tier 2 and 3 supports to respond to student needs in this peri-COVID era [33]. 

Several important facilitators and barriers were distilled in this study that help to inform future “grow your own” models. Similarly to prior studies, the cadre members lived and worked in the community, which motivated them to engage in this degree path [7]. Whereas, unlike other programs training school psychology professionals [6], recruitment challenges and decisions related to the financial burden of the program were not evident factors influencing its implementation. Retention efforts were facilitated through financial incentives, specifically through knowing from the beginning of the recruitment to the enrollment process that the prospective participants’ tuition was likely to be a key facilitator for engaging students in the program and beginning the pursuit of their MSW degrees [7,8]. Once enrolled, the cadre members discussed the course content and strategies utilized in the classroom as two of the most prominent facilitators of learning. The cadre members also described experiencing diverse classrooms, where group processing, modeling, experiential activities, case studies, panels and presentations, and application-based tasks helped them learn about the social work profession, mental health, differential diagnoses, SSW, diversity and cultural competence, social justice, and processes such as needs assessment, program evaluation, and advocacy-based practice. Many of these outcomes align with practices outlined in the National SSW Practice model [24], demonstrating a progression of skills supportive of meeting behavioral health needs in schools.

The cadre members also shared how the organization of the program, with its hybrid design of in-person and online courses, along with some opportunities for choice in progression toward their degree, also supported their learning and retention. The courses offered in the district immediately after school also were presented as facilitators for accessing in-person courses, as the participants could finish the school day and then travel a brief distance to attend courses. In other programs, distance from the university has been considered a unique challenge, one that inhibits access to faculty, materials, and university resources [6]. Here, the strength of the district–university partnership was described by the cadre as “compassionate, supportive, invested, connected, intentional, and thoughtful”. For those in the study, these qualities helped navigate unique learning and participation barriers and organize the program in a way that mitigated the risk of poor attendance, dropout, and heightened stress. In the future, social work and other training programs might co-locate courses for various students, especially if their shared lived experiences facilitate learning, as seen in this study.

The make-up of the instructors and cadre, as well as the first practicum experience, also served as facilitators of learning. The OSU-CAYCI team specifically identified instructors to support this cadre, noting the need to identify those with experience with adult learners and/or extensive expertise in each subject area. The instructors were described as highly personable, adaptive to the cadre’s unique learning situations (e.g., working full-time, etc.), and able to elevate and enhance the cadre’s past experiences to advance their knowledge and skills. Furthermore, the cadre members learned from one another and provided social and academic support that helped them feel a part of something bigger than themselves. Additionally, communication among the field program director, the faculty leads of the project, and the adjunct instructors was facilitated through the OSU-CAYCI liaison, which helped to streamline communication and scaffold learning. All of these factors were facilitators that were described by cadre members.

In addition, the supervision structure and experiences cultivated through the first practicum experience were intentional, and the cadre members were pushed to delve into different roles outside of a school setting to support a population of highly vulnerable youth. Having a former SSWer on the team to help contextualize learning in the field in courses and practice inevitably helped the students make connections. Furthermore, this person allowed them the space to process feelings associated with role transitions, cultural humility, diversity, and broader social, economic, and health disparities in the community. Pushing students to learn in new settings outside of their comfort zones and matching practicum instructors to students’ future areas of practice may help facilitate learning among SSW trainees. However, universities may not have the same capacity as OSU-CAYCI to engage 25 students in similar field experiences, despite the potential for this cohort design to support camaraderie and shared learning.

The design of the program also presented specific barriers to learning. For example, the cadre members shared the intensity and demands that influenced their time with family and personal lives. There were opportunities to improve communication about the progression of the programming, along with scheduling processes through the university. Further, many of the cadre members were not satisfied with the online courses, noting that there was a significant amount of busy work and outdated content that was not perceived as supporting their learning and growth. More clinical content (the focal area of year two) was also desired. OSU program leaders have opportunities to strengthen these areas to improve the student experience and support future retention efforts. Ultimately, the experiences shared by the cadre at the mid-way point helped to distill factors that facilitated several key learning outcomes. The cadre members with backgrounds in education reported a greater understanding of the profession of social work, gaining self- and social awareness, and obtaining foundational therapeutic skills. Importantly, these outcomes align with the knowledge, skills, and competencies outlined by the Council on Social Work Education [38].

At a broader level, there are key nuances to discuss regarding this “grow your own” program that may influence replication and future partnership models. Many districts may face obstacles with funding, given that COVID-19 relief dollars were used in this program and that these are not being renewed or sustained in this peri-COVID era. The DCS committed not only these relief funds to this project, but also dollars to support the employment of 25 full-time SSW interns as the cadre moves into their second year of the program. Subsequently, in the following fiscal year, the district committed dollars toward 25 full-time mental health specialist roles while also balancing other fiscal changes (e.g., shifts to an all-day kindergarten and opening a new elementary school). These decisions reflect a financial commitment to mental health that is unparalleled with those seen in other districts, states, and even nationally. However, many districts may face challenges in being able to pay tuition in the short term and then ultimately creating new roles post-graduation for trainees while hiring new teachers to fill open positions to replicate this model.

For replication at the policy level, a multi-district collaboration is a potentially transferrable model that could be fiscally feasible and help districts to build their capacity to support behavioral health needs. In this model, multiple districts could work collaboratively to identify one to two members of their current school staff and utilize financial resources to enroll them in a “grow your own” program in partnership with a local university. The financial investments from districts could therein be smaller, but they could help to cultivate a shared community of practice in a region that ultimately improves service delivery and generates a pipeline to address behavioral health workforce shortages. Alternatively, federal or state education departments may revamp workforce development programs (e.g., Mental Health Service Demonstration grants [21,22]) to fund local education agencies to target current employees to address behavioral health workforce shortages specifically. Simultaneously, grants could include financial incentives to support recruiting new teachers to fill gaps and generate a pipeline of resources that help schools address shortages using committed in-house personnel. The “grow your own” program outlined here demonstrates how a pre-established relationship and commitment among trainees to a district may support retention efforts and translate into the successful sustainment of behavioral health staff in schools.

### 6.1. Limitations

This study has several limitations. First, school administrators shared the district survey, and not all perspectives of staff in the district were captured. The data may, therefore, be skewed, and they may not be representative of the overall perceptions of behavioral health needs. Similarly, quantitative data represent staff perspectives of school and student needs. However, student perspectives and clinical measures may be needed to substantiate behavioral health needs accurately in this district. Additionally, the cadre members interviewed were notably those highly interested in career changes and in pursuing an MSW during their enrollment in the study, meaning social desirability may have played a role in the feedback and learning outcomes shared. Several chose not to participate and may have been less satisfied with the program overall. Further, the transparency of the district–university partnership, while supportive of the project overall, may have biased students into sharing positive perceptions of the program because of the district’s role in supporting their tuition. The experiential backgrounds of these students, including paraprofessionals, licensed teachers, school counselors, and school psychologists, may have influenced their perceptions of knowledge, skills, and competencies. The training and practice differences associated with each student’s current professional role likely influenced how they answered the questions and may continue to shape their experiences in the program’s second year overall. Finally, the data were cross-sectional and only condensed to one district, limiting generalizability to other contexts, districts, or educational programs.

### 6.2. Future Research

This study is part of a five-year research study underway to examine the short- and long-term influence of this “grow your own” model on cadre members’ learning outcomes and one school district’s capacity to deliver mental health services amidst ongoing behavioral health workforce shortages. As the cadre members continue to progress through the program, additional data will be captured to document the outputs of this policy pilot further. For example, the cadre members completed pre-program surveys assessing their perceptions of the nine social work competencies outlined by the Council on Social Work Education [38]. Upon graduating, the cadre members will again complete this measure, allowing researchers to explore perceptions of learning throughout the entire program. Additionally, interviews and focus groups will also take place with cadre members, adjunct instructors, field supervisors, and district leaders to distill their perceptions of facilitators or barriers, along with the behavioral changes (including questions such as did you do anything differently? What happened as a result, etc.?) associated with participation in or affiliation with the program.

From a policy perspective, secondary data from the MSW program will be compared to competency-based measures of this cadre and other social work students to see how the design of this program can inform the future delivery of MSW degrees for practitioners with an interest in working in schools. The researchers also will track long-term outcomes, including job retention, satisfaction, and performance, as well as changes over time using another district-level staff survey on their perceptions of behavioral health needs. These approaches will help to distill how this investment using COVID-19 relief and other funds supported broader employment and performance outcomes, and how it can inform policy initiatives from local, state, and federal sources to address behavioral health workforce shortages in schools.

## 7. Conclusions

There is a nationwide shortage of behavioral health providers prepared to work in schools. This study significantly contributes to the field by being the first known “grow your own” program designed to train and prepare mid-career district staff to become SSWers and obtain mental health specialist jobs in schools, resulting in an innovative approach to addressing behavioral health workforce shortages by leveraging district resources and partnerships with universities. The study showcases how such a model can be implemented effectively and provides valuable insights into the successes, challenges, and systems-level innovations that can guide future replication. The program’s retention rate and participant feedback underscore its potential as a sustainable and scalable solution for districts facing similar workforce shortages, particularly in the peri-COVID era. The study offers practical recommendations for policy-level adoption, including multi-district collaborations and funding strategies, making this “grow your own” approach a vital resource for educational and behavioral health stakeholders.

## Figures and Tables

**Table 1 behavsci-14-00813-t001:** Coursework and Practicum Design (Five Semesters).

Design	Semester				
Program Component	Summer	Fall	Spring	Fall	Spring
Credit hours (courses) enrolled	9 credit hours (3 courses and first practicum)	13 credit hours (4 courses)	12 credit hours (4 courses)	16 credit hours (4 courses and second practicum)	13 credit hours (3 courses and second practicum)
Course delivery mechanism	3 online, asynchronous courses	Two online, asynchronous courses and two in-person courses on DCS campus	Two online, asynchronous courses and two in-person courses on DCS campus	Optional: Two or three online, asynchronous courses and one or two in-person courses on DCS campus	Two online, asynchronous courses and one in-person course on DCS campus
Practicum experience	LiFEsports at The Ohio State University			DCS elementary, middle, or high school with one day a week spent shadowing a partner cadre member in a different school	
Working full-time (Yes or No)	No	Yes	Yes	No	No

**Table 2 behavsci-14-00813-t002:** District-Wide Staff Perceptions of Behavioral Health Needs (N = 596).

In This Building, There Are Enough Resources to:	% Do Not Have	% Have and Do Not Need More	% Have and Need More
Tier I			
Screen students for mental health concerns	17%	18%	65%
Provide mental health literacy programming	34%	13%	53%
Support classroom prevention planning	25%	16%	59%
Tier II			
Conduct home visits to support families	50%	15%	35%
Engage in consultation for teachers	30%	15%	55%
Implement small group interventions	15%	15%	70%
Tier III			
Provide case management services	25%	15%	60%
Provide individual counseling	9%	11%	80%
Support crisis intervention	9%	16%	75%

**Table 3 behavsci-14-00813-t003:** Facilitators of Learning (N = 19).

Theme and Subthemes	%
Effective Classroom Strategies	100%
Practical application: “I can see myself using this knowledge and information right away”	68%
Panels and presentations from social workers in the community; learning about different agencies and roles of social workers in the community	53%
Space for group processing and open conversations (i.e., small groups, reflection opportunities)	50%
Ability to model after others by sitting in on meetings and watching others in helping roles	37%
Experiential activities in classes (i.e., role play)	37%
Learning activities requiring identification and utilization of EBPs	37%
Case studies/hearing of personal stories/scenarios/ethical dilemmas	26%
Specific Learning Experiences and Content	100%
Relevance of the content of the class on SSW to context and future positions (i.e., positive behavioral intervention supports, multi-tier systems of support, FERPA/HIPPA)	79%
Learning about mental health, particularly diagnosis	74%
Diversity, cultural competence, and social justice content and opportunities that were safe to share and learn	68%
Space for real conversations about race, privilege, and social justice	68%
Learning about community resources	63%
Learning about social work as a profession	58%
Scaffolding of course content, especially in relation to social justice	50%
Broader needs assessment, program design, implementation and evaluation	26%
Organization of the Program/Class Structure	89%
In-person classes	74%
Flexibility and autonomy in course selections	58%
Timing of classes (right after school) and location (hosted at DCS)	50%
Online setup was convenient and afforded flexibility	16%
Online classes broadened learning as other students from outside of the cadre were in the class	16%
Instructional Techniques and Instructor Qualities	84%
Knowledge and expertise	58%
Flexibility in attendance, grading, and allowing for virtual attendance	58%
Calm, relatable, authentic, reflective, and served as good role models of therapeutic skills	53%
Accessibility and helpfulness	42%
Effective teachers in the classroom	32%
Ability to resolve tension and manage a cadre of adult learners	21%
Cadre/Cohort Learning	84%
Cadre as a whole, camaraderie, and support (i.e., “everyone is in the same boat”)	84%
Opportunity to “tap into everyone’s lived experiences and strengths”	37%
University–District Partnership	84%
Leaders at OSU were supportive and provided structure and support	95%
Support from DCS	63%
Communication of expectations and what was coming up next	26%
Rigor/well-rounded program	37%
Focus on self-care	26%
Intentionality and thoughtfulness of the application and learning process	16%
LiFEsports Field Placement	68%
Setting was different from those in traditional education and DCS	53%
Allowed for better understanding of issues facing youth today and diverse backgrounds of youth	50%
Supervision and structure provided during LiFEsports placement	16%
Other Facilitators	89%
Pre-requisite knowledge base (i.e., good at making relationships, working with youth)	74%
Current self-help practices	37%
Time management and organizational skills	32%
Self as representative of a minoritized group and/or with family members	26%
Supportive others (i.e., family)	16%

**Table 4 behavsci-14-00813-t004:** Learning Outcomes (N = 19).

Theme and Subthemes
Knowledge of Generalist Social Work Practice
Enhanced understanding of macro/mezzo forces and systems perspective
Learned about the SSW subdiscipline and specific practices
Gained knowledge of the profession of social work
Described confidence in helping others and being a social worker
Articulated skills in identifying community resources and how to find them
Voiced knowledge of EBPs, how to find them, and choosing strategies/interventions
Discussed knowledge of how to diagnose and conduct needs assessments
Recognized the importance of advocacy and growing confidence in advocacy skills
Reported understanding of the importance of working with families
Self- and Social Awareness
Appraised own biases and privilege and reported being less judgmental
Discussed learning to be more open-minded/broadened perspective of the world
Described ability to interpret ethics and handle dilemmas
Ongoing Lessons as a Social Worker and Lifelong Learner
Outlined the importance of self-care
Commented on learning skills such as “how to be a student again”
Realized social workers are growing in skills and need reflection/consultation
Displayed awareness of how the social work lens is entirely different from an educator lens
Therapeutic Skills
Articulated need to go to where the client is (i.e., “check self at the door”, “be curious”)
Exemplified listening and communication skills (i.e., be “comfortable with the silence)
Discussed the development of patience
Reflected on growth in skills to build rapport and relationship-building skills
Described understanding of the need to honor who others are and their lived experiences
Defined and described self-determination
Articulated understanding the importance of focusing on strengths and the good in others

Note. “EBPs” stand for evidence-based practices.

## Data Availability

The datasets presented in this article are not readily available because they are part of an ongoing study. Requests to access the datasets should be directed to the corresponding author.

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
