# Peer review of "Grow Your Own School Mental Health Specialists: A Policy Pilot to Address Behavioral Health Workforce Shortages in Schools"

_behavsci, 2024, doi:10.3390/bs14090813_

Round 1

Reviewer 1 Report

Comments and Suggestions for Authors

Introduction

The introduction effectively sets the stage for the paper by clearly outlining the problem of behavioral health workforce shortages in schools and the innovative approach taken by the school district and university partnership to address this issue. However, it could benefit from a more explicit statement of the research objectives and how they align with the broader literature on "grow your own" programs. Including a brief overview of the paper's structure at the end of the introduction would also guide readers through the following sections.

Literature Review

The literature review is thorough and well-structured, providing a solid background on the issue of behavioral health needs in schools and the concept of "grow your own" programs. The references are current and relevant, with a strong emphasis on the need for more social workers in educational settings. However, the review could be strengthened by more critically engaging with the literature, discussing potential gaps, and highlighting how this study addresses those gaps. Additionally, more connections between the literature and the specific context of the study would make the review more cohesive.

Methodology

The methodology section is detailed and provides a clear explanation of how participants were recruited, selected, and what the program entailed. The use of both qualitative and quantitative data collection methods is a strength, as it allows for a more comprehensive understanding of the program's impact. However, the section could be improved by providing more information about the specific tools used for data analysis and the rationale behind choosing them. Additionally, discussing any potential biases or limitations in the study design would provide a more critical perspective.

Research Questions

The research questions are implicitly addressed throughout the paper, but they are not clearly stated in a separate section. Explicitly stating the research questions early in the paper would help focus the reader and clarify the study's aims. This would also make it easier to connect the findings to the questions, providing a clearer narrative thread throughout the paper.

Results

The results section is comprehensive and well-organized, with tables and figures that effectively illustrate the findings. The use of direct quotes from participants adds depth to the qualitative analysis, and the quantitative data is presented clearly. However, the section could be enhanced by more explicitly linking the findings to the research questions and discussing how they contribute to the existing body of knowledge. A more detailed discussion of the statistical analysis methods and their significance would also strengthen this section.

Implications

The paper does a good job of discussing the practical implications of the study for school districts and policymakers. The innovative "grow your own" model has clear potential for addressing workforce shortages, and the discussion of its applicability to other contexts is insightful. However, the implications could be expanded by considering potential challenges in replicating the model, such as funding constraints or differences in district needs. Additionally, the discussion could benefit from a more explicit connection to the broader literature on workforce development and behavioral health.

Conclusions

The conclusion effectively summarizes the key findings and emphasizes the importance of the study. However, it could be strengthened by offering more concrete recommendations for future research or practice. Discussing the study's limitations, if not discussed in the methodology, in this section would also provide a more balanced view and help readers understand the scope of the findings.

Tables and Figures

The tables and figures used in the paper are relevant and enhance the reader's understanding of the data. However, the captions could be more descriptive, providing a brief explanation of what the reader should take away from each table or figure. Additionally, integrating these visuals more closely with the text by directly referencing them in the discussion of results would make the paper more cohesive.

Overall

The overall structure of the paper is logical and flows well from one section to the next. A stronger emphasis on the study's contributions to the field, particularly in the conclusion, would leave the reader with a more lasting impression of the paper's significance.

Comments on the Quality of English Language

The paper is generally well-written, with clear and concise language. However, there are some areas where the writing could be tightened to improve clarity and readability. For example, there are a few instances of passive voice that could be revised for a more active and engaging tone.

Author Response

Reviewer 1

Introduction

The introduction effectively sets the stage for the paper by clearly outlining the problem of behavioral health workforce shortages in schools and the innovative approach taken by the school district and university partnership to address this issue. However, it could benefit from a more explicit statement of the research objectives and how they align with the broader literature on "grow your own" programs. Including a brief overview of the paper's structure at the end of the introduction would also guide readers through the following sections.

  • Thank you for this comment. The research questions are now explicitly stated in the “Current Study” section. A brief overview of the paper’s structure is now included at the end of the introduction to guide the reader through the subsequent sections.

Literature Review

The literature review is thorough and well-structured, providing a solid background on the issue of behavioral health needs in schools and the concept of "grow your own" programs. The references are current and relevant, with a strong emphasis on the need for more social workers in educational settings. However, the review could be strengthened by more critically engaging with the literature, discussing potential gaps, and highlighting how this study addresses those gaps. Additionally, more connections between the literature and the specific context of the study would make the review more cohesive.

  • These comments helped us return to the literature to critically appraise what has been done and highlight gaps and prior challenges noted by authors or other studies, including recruitment and retention barriers, especially if costs or travel were not accounted for in the program design. We also noted how behavioral health approaches are often implemented, initiated, or designed at federal or state levels. These sentiments were added to the literature review and helped shape additional revisions to the “Current Study” section to further contextualize the study and show our unique contribution to the literature.

Methodology

The methodology section is detailed and provides a clear explanation of how participants were recruited, selected, and what the program entailed. The use of both qualitative and quantitative data collection methods is a strength, as it allows for a more comprehensive understanding of the program's impact. However, the section could be improved by providing more information about the specific tools used for data analysis and the rationale behind choosing them. Additionally, discussing any potential biases or limitations in the study design would provide a more critical perspective.

  • SPSS is included as the analysis tool for quantitative data. We did not include more details about this tool or rationale behind its selection, as it is commonly used and cited in research literature. For qualitative analyses, we better articulated using Excel to engage in content analysis and described why the content analysis was used to analyze the qualitative data. Biases and design limitations are also discussed in the limitations section.

Research Questions

The research questions are implicitly addressed throughout the paper, but they are not clearly stated in a separate section. Explicitly stating the research questions early in the paper would help focus the reader and clarify the study's aims. This would also make it easier to connect the findings to the questions, providing a clearer narrative thread throughout the paper.

  • Thank you for this comment. Reviewer 2 also made this suggestion. Research questions have been explicitly added to the “Current Study” section.

Results

The results section is comprehensive and well-organized, with tables and figures that effectively illustrate the findings. The use of direct quotes from participants adds depth to the qualitative analysis, and the quantitative data is presented clearly. However, the section could be enhanced by more explicitly linking the findings to the research questions and discussing how they contribute to the existing body of knowledge. A more detailed discussion of the statistical analysis methods and their significance would also strengthen this section.

  • Thank you for this comment. We have revised each section of the results to restate the research question before delving into the specific findings. We have included a discussion of how the results relate to the existing body of literature and expanded the discussion of the analysis method in the limitations sections of the manuscript.

Implications

The paper does a good job of discussing the practical implications of the study for school districts and policymakers. The innovative "grow your own" model has clear potential for addressing workforce shortages, and the discussion of its applicability to other contexts is insightful. However, the implications could be expanded by considering potential challenges in replicating the model, such as funding constraints or differences in district needs. Additionally, the discussion could benefit from a more explicit connection to the broader literature on workforce development and behavioral health.

  • The final two paragraphs of the discussion now explicitly link back to broader literature and also expand upon potential funding constraints and potential challenges in replication.

Conclusions

The conclusion effectively summarizes the key findings and emphasizes the importance of the study. However, it could be strengthened by offering more concrete recommendations for future research or practice. Discussing the study's limitations, if not discussed in the methodology, in this section would also provide a more balanced view and help readers understand the scope of the findings.

  • The conclusion section has been revised to address these comments and the discussion section was also revised to include more concrete considerations for research and practice.

English Usage and Mechanics

The paper is generally well-written, with clear and concise language. However, there are some areas where the writing could be tightened to improve clarity and readability. For example, there are a few instances of passive voice that could be revised for a more active and engaging tone.

  • Thank you. When identified, sentences with passive voice were revised to active voice.

Tables and Figures

The tables and figures used in the paper are relevant and enhance the reader's understanding of the data. However, the captions could be more descriptive, providing a brief explanation of what the reader should take away from each table or figure. Additionally, integrating these visuals more closely with the text by directly referencing them in the discussion of results would make the paper more cohesive.

  • Thank you for this comment. The results section seeks to broadly summarize each table, and we have included references back to the tables in the discussion to improve readability.

Overall

The paper's overall structure is logical and flows well from one section to the next. A stronger emphasis on the study's contributions to the field, particularly in the conclusion, would leave the reader with a more lasting impression of the paper's significance.

  • Thank you for this comment. The conclusion section has been revised to leave a more lasting impression on the reader about the contribution of this article to the literature.

Reviewer 2 Report

Comments and Suggestions for Authors

Dear author/s

Thank you for allowing me to review the article:

Grow Your Own School Mental Health Specialists: A Policy  Pilot to Address Behavioral Health Workforce Shortages in  Schools

The subject of the article is very important to the educational world and presents a new angle of view that is not necessarily known to everyone and not all educators recognize the importance of mental health.

Abstract: In this section, two more sentences should be added describing the findings and the key insights from the study.

Keywords: Rethinking is required for the correct choice of the keywords for example "school behavioral health" appears only once and only in this section.

In addition, I recommend the use of initials for example:  "school social work" = "SSW"

Introduction:  The introduction and the literature review should be organized as two separate chapters as is customary in a scientific article.

In lines 53-65, there is no reference to the sources of the information.

At the beginning of the literature review chapter, precise definitions of health concepts that appear in the article should be written for the benefit of the future reader.

In line 84 - it is written  " settings [13] (p. 12)."  without a quote.

 lines 142-145 - an appropriate reference for information should be added.

Current Study- lines 147-154 - Demographic data of the research participants should be added, such as gender, age, education, etc.

District-University Partnership: In this section, the advantages of the partnership between the university and the schools should be added, it is also possible to rely here on other partnerships conducted between an academic institution and the schools, as for example in the chapter on the practice of teaching students and the advantages inherent in this.

Table number 1: should appear in line 246 after the subchapter of the "Program Design"

General note: All tables are not adapted to the page size.

Methods: Clearly written - no comments.

Table number 4: must be organized visually to be easy to read

The discussion chapter: requires re-editing and reference to suitable biographical sources from the literature review. See for example in lines: 456-472 there is no reference to literature as well as in lines 499-549.

References:  The list of references is not written in a uniform manner and according to the required rules of the journal.

For example, in item number 1 the year is highlighted and in item number 3 the year is not highlighted, in addition, the links are not active.

I hope that the authors of the article will find my comments useful in improving their article.

I look forward to hearing from the author/s regarding the resubmission.

Good luck.

Author Response

Reviewer 2

The subject of the article is very important to the educational world and presents a new angle of view that is not necessarily known to everyone and not all educators recognize the importance of mental health.

Abstract: In this section, two more sentences should be added describing the findings and the key insights from the study.

  • The following sentences were added to the abstract: “Findings from a district-wide staff survey indicated a high level of need for individual counseling, crisis intervention, and small group interventions. Additionally, qualitative interviews revealed that learning among program participants was facilitated by effective classroom strategies and specific learning experiences integral to the program’s design. These facilitators supported key learning outcomes, including general social work knowledge, self-awareness, and therapeutic skills.”

Keywords: Rethinking is required for the correct choice of the keywords for example "school behavioral health" appears only once and only in this section. In addition, I recommend the use of initials for example:  "school social work" = "SSW"

  • Thank you. This keyword was removed and changed to “grow your own.” School social work was also abbreviated as SSW and school social workers to “SSWers” throughout the article.

Introduction:  The introduction and the literature review should be organized as two separate chapters as is customary in a scientific article.

  • We’ve added a header to outline the literature review separate from the introduction.

In lines 53-65, there is no reference to the sources of the information.

  • Thank you. Several references regarding Grow Your Own programs (reviews and recent studies), along with references about the credentialing and competencies of school social workers, were added to this section. [See 9,10,11,12].

At the beginning of the literature review chapter, precise definitions of health concepts that appear in the article should be written for the benefit of the future reader.

  • Behavioral health is now broadly defined for the reader at the onset of the literature review. Because our focus on behavioral health in schools remains broad throughout the article, we opted to focus on this definition rather than specific diagnoses such as anxiety and depression. The literature review also includes citations supportive of this definition.

In line 84 - it is written  " settings [13] (p. 12)."  without a quote.

  • Thank you - “Pg. 12” on line 84 was removed as this sentence does not include a quote.

Lines 142-145 - an appropriate reference for information should be added.

  • References were added.

Current Study- lines 147-154 - Demographic data of the research participants should be added, such as gender, age, education, etc.

  • Demographic data was added to the “Recruitment and Sample” section.

District-University Partnership: In this section, the advantages of the partnership between the university and the schools should be added, it is also possible to rely here on other partnerships conducted between an academic institution and the schools, as for example in the chapter on the practice of teaching students and the advantages inherent in this.

  • A sentence on the strengths of having a pre-established relationship between the university and school district is now explicitly stated in this section. The remainder of this ask from the reviewer was unclear regarding the chapter. If clarity can be provided, we’d be happy to address it.

Table number 1: should appear in line 246 after the subchapter of the "Program Design"

General note: All tables are not adapted to the page size.

  • Table 1 was moved under the “Program Design” section. Tables were adapted to page size. We hope the copyeditor will further help align tables to page requirements.

Methods: Clearly written - no comments.

Table number 4: must be organized visually to be easy to read

  • Table 4 was revised to improve readability and better describe the outcomes discussed by interviewees.

The discussion chapter: Requires re-editing and reference to suitable biographical sources from the literature review. See for example in lines: 456-472 there is no reference to literature as well as in lines 499-549.

  • We included references from the literature review in the discussion. Of note, some aspects of the discussion needed to unpack findings a bit more given few “grow” your own studies have specifically examined learning outcomes and nuanced barriers and facilitators using qualitative approaches.

References:  The list of references is not written in a uniform manner and according to the required rules of the journal. For example, in item number 1 the year is highlighted and in item number 3 the year is not highlighted, in addition, the links are not active.

  • Thank you, the references were cleaned and all links are now active.

I hope that the authors of the article will find my comments useful in improving their article. I look forward to hearing from the author/s regarding the resubmission. 

Round 2

Reviewer 2 Report

Comments and Suggestions for Authors

Thanks for submitting the revised version of the article. The authors of the article correct according to the reviewer's instructions, but it should be noted that many bibliographic references exceed 20% of the total of all article are self-citations.

Despite the importance of the self-citations, I ask the authors of the article to reduce the self-citations, especially of the first and second authors (Samantha Bates Dawn Anderson-Butcher)

After these corrections are made, the article will be ready for publication.

Thanks to the authors of the article for their cooperation

Greetings,

The reviewer

Author Response

Comment 1: 

Thanks for submitting the revised version of the article. The authors of the article correct according to the reviewer's instructions, but it should be noted that many bibliographic references exceed 20% of the total of all article are self-citations.

Despite the importance of the self-citations, I ask the authors of the article to reduce the self-citations, especially of the first and second authors (Samantha Bates Dawn Anderson-Butcher)

Response: Article self-citations have been addressed and now represent less than 20% of those in the references. We removed 3 specifically that could be articulated/captured by other citations. Thank you.